# Mutual Information between EDA and EEG in Multiple Cognitive Tasks and Sleep Deprivation Conditions

**DOI:** 10.3390/bs13090707

**Published:** 2023-08-25

**Authors:** David Alejandro Martínez Vásquez, Hugo F. Posada-Quintero, Diego Mauricio Rivera Pinzón

**Affiliations:** 1Electronic Engineering Faculty, Universidad Santo Tomás, Bogotá 110231, Colombia; 2Department of Technology, Universidad Pedagógica Nacional, Bogotá 110221, Colombia; dmrivera@pedagogica.edu.co; 3Department of Biomedical Engineering, The University of Connecticut, Storrs, CT 06269, USA; hugo.posada-quintero@uconn.edu

**Keywords:** EEG (electroencephalography), EDA (electrodermal activity), ECG (electrocardiography), HRV (heart rate variability), information theory, mutual information, sleep deprivation

## Abstract

Sleep deprivation, a widespread phenomenon that affects one-third of normal American adults, induces adverse changes in physical and cognitive performance, which in turn increases the occurrence of accidents. Sleep deprivation is known to increase resting blood pressure and decrease muscle sympathetic nerve activity. Monitoring changes in the interplay between the central and autonomic sympathetic nervous system can be a potential indicator of human’s readiness to perform tasks that involve a certain level of cognitive load (e.g., driving). The electroencephalogram (EEG) is the standard to assess the brain’s activity. The electrodermal activity (EDA) is a reflection of the general state of arousal regulated by the activation of the sympathetic nervous system through sweat gland stimulation. In this work, we calculated the mutual information between EDA and EEG recordings in order to consider linear and non-linear interactions and provide an insight of the relationship between brain activity and peripheral autonomic sympathetic activity. We analyzed EEG and EDA data from ten participants performing four cognitive tasks every two hours during 24 h (12 trials). We decomposed EEG data into delta, theta, alpha, beta, and gamma spectral components, and EDA into tonic and phasic components. The results demonstrate high values of mutual information between the EDA and delta component of EEG, mainly in working memory tasks. Additionally, we found an increase in the theta component of EEG in the presence of fatigue caused by sleep deprivation, the alpha component in tasks demanding inhibition and attention, and the delta component in working memory tasks. In terms of the location of brain activity, most of the tasks report high mutual information in frontal regions in the initial trials, with a trend to decrease and become uniform for all the nine analyzed EEG channels as a consequence of the sleep deprivation effect. Our results evidence the interplay between central and sympathetic nervous activity and can be used to mitigate the consequences of sleep deprivation.

## 1. Introduction

Human brain activity has been widely studied through the effects that determine cognitive tasks produced in electroencephalography (EEG) signals, specifically in alpha, beta, delta, theta and gamma oscillations. In this regard, tasks requiring the storing and manipulation of information, commonly known as working memory tasks, have demonstrated an increase in delta oscillations associated to the concentration that individuals require to perform them [1]. On the other hand, tasks demanding timing and inhibition, in which individuals perceive visual or auditory stimuli and react in an inhibitory or encouraged form, have been associated to the power increase in alpha waves [2], which, according to [3], tend to diminish with drowsiness and sleep. In contrast, the power increase in theta waves has been related to the fatigue increment caused by sleep deprivation when tasks requiring attention and reaction times are performed [4,5], which demonstrates that these kinds of oscillations are useful to determine tiredness conditions that could be dangerous in occupational contexts demanding rapid response times such as military, health care, and others that in addition, require long periods of concentration or nocturnal activity. The effects of sleep deprivation have also been studied through many approaches that use, in most of the cases, tasks involving attention and vigilance to analyze the EEG components directly [6,7]. In addition, the relationship between EEG bands and EDA components is studied in some approaches to find correlations between phasic and theta waves and determine connections between brain activity and sympathetic arousal [8]. The relationship between EDA and HRV is also considered to find correlations between peripheral sympathetic arousal and deterioration in cognitive abilities [9], or to identify risks due to attention failure in individuals [10]. Signals’ interaction has not only been used for sleep deprivation analysis, but also in other medical contexts such as the adolescent attention deficit hyperactivity disorder described in [11], where a negative correlation between theta and EDA, particularly for non-specific skin conductance responses (NS.SCRs), is identified in attention deficit hyperactivity disorder patients. A high correlation between EDA, specifically the skin conductance level (SCL, also known as tonic component) and the EEG alpha and beta bands, is found in [12]. This result is introduced in a substantive relationship between cerebral function and autonomic arousal. A similar study, but this time including a combination of ECG, EDA and EEG signals, was conducted in [13], providing some evidence on the benefits of Tibetan Singing Bowls in metastatic cancer patients. The previously mentioned works have put special interest in the EDA signal, considered an index of autonomic sympathetic activity [8], and its relationship with brain and heart activities in order to identify cognition stress and emotional or mental states when individuals cannot self-report them [14].

Statistically, these signal relationships are commonly measured using tools such as ANOVA, Pearson’s correlation coefficient, Friedman test and Spearman correlation, among others that, in spite of their relevance, are limited to the linear scope, i.e., they are focused on determining how a variable increases or decreases in the function of the other, which could lead us to ignore relevant information hidden in the non-linearities. An important tool to face this problem is the information theory, which, through the mutual information concept, can identify how much information a signal provides about another, considering both linear and non-linear interactions [15]. In this regard, we calculate the mutual information between EEG and EDA to provide new insights about the connection between brain activity and the autonomic sympathetic activity in different cognitive contexts that, to the best of our knowledge, have been widely studied in only linear interaction analyses.

This paper is divided as follows. Section 2.1 describes the main concepts related to information theory, EDA and EEG signals and the tasks used in this analysis. Section 2 describes the implemented methods to collect the EDA and EEG information, and the form in which mutual information is calculated. Section 3 describes the simulation results that we finally analyze in Section 4 and Section 6.

## 2. Materials and Methods

In this study, whose protocol was approved by the Institutional Review Board of The University of Connecticut, ten healthy (without sleep disorders) volunteers (7 male) between the ages of 25–35 were involved. The participants had to perform the tasks described in Section 2.2 every two hours during a 25 h time period.

Data acquisition started at 10 a.m., with EAT being the first executed task, followed by the ship search, N-Back, and PVT tasks. The analyzed data for each case included the EEG signals, covering the five frequency bands, as well as the EDA signal and including its phasic and tonic components. Additionally, ECG signals were collected during each trial to analyze the effect of sleep deprivation in HRV. More details in this regard are given in Section 2.3. All participants were requested to reach the experimental site within two hours after waking up, and they had to fill a questionnaire to confirm the quality and amount of the sleep they had before the experiment. During the study, an HP 78354A ECG monitor manufactured by Hewlett–Packard, Palo Alto, CA, USA, was employed to gather ECG data, while a galvanic skin response amplifier, specifically the FE116 model from ADinstruments, Colorado Springs, CO, USA, was utilized to collect EDA (Electrodermal Activity) data. Prior to each EDA recording trial, the device was calibrated to zero. To gather the EEG signal, we used an actiCHamp amplifier (Brain Products GmbH, Gilching, Germany) with an EasyCap electrode system (EasyCap GmbH, Herrsching-Breitbrunn, Germany).

### 2.1. Preliminaries

#### 2.1.1. Information Theory

One of the essential concepts in information theory is the entropy, which determines the uncertainty level of a random variable *X* from its probability distribution p(x). Mathematically, entropy is given by
(1)$$H\left(X\right)=-\sum_{x\in X}p(x)\log p(x)\;\left[bits\right],$$
where p(x)=Pr{X=x}, x∈X, with X as the alphabet of *X*. The maximum entropy of *X* occurs when each x∈X has the same probability, and it is reduced when prior probability information is given [16].

The entropy concept can be extended to multiple random variables, which let us define joint entropy and conditional entropy for dependent variables. In this sense, the joint entropy of *X* and *Y* is described as
(2)$$\begin{array}{cc} H(X,Y) & =-\sum_{x\in X}\sum_{y\in Y}p(x,y)\log p(x,y)\\ & =H(X)+H(Y|X). \end{array}$$

An explicit relationship between the entropy measures is shown in the Venn diagram of Figure 1. Observe how, in addition to the joint, conditional and independent entropies, we have a new measure given by the intersection between H(X) and H(Y), which is known as the mutual information and represents the information that any random variable has about the other, in other words, it describes the uncertainty reduction about a random variable for having information about the other. This is described by the expression
(3)$$\begin{array}{cc} I(X;Y) & =\sum_{x\in X}\sum_{y\in Y}p\left(x,y\right)p\left(x\right)\log\frac{p(x,y)}{p(x)p(y)}\\ & =H(X)-H(X|Y). \end{array}$$

Mutual information has gained special attention in recent years in areas such as multi-agent systems [17], coverage control [18,19], distributed control in micro-grids [20], and neuroscience [15,21] due to its capability to analyze relationships between data from different contexts (e.g., voltage values, stimulus light position, animal position), the ability to detect linear and nonlinear interactions, and its usage in multivariate systems.

#### 2.1.2. Electrodermal Activity (EDA)

Electrodermal Activity (EDA) is the acronym used to describe the change in skin conductance that reflects the sympathetic nerve activity on sweat glands. This is composed mainly by the tonic component (also known as Skin Conductance Level: SCL) and the phasic component (also known as Skin Conductance Response: SCR). The tonic component (SCL) refers to the long-term fluctuations in the EDA signal that are not related to particular external stimulus, but to particular individual emotions, thoughts or electro-dermal instability. On the other hand, the phasic component (SCR) is associated to the response of particular external stimulus such as complex cognitive tasks, association between loud tones, images or shapes with threatening events, among others, that are applied to individuals at determined time intervals within 1–5 s [22,23]. The responses associated to this kind of stimulus are commonly known as Event-Related SCR (ER-SCR), whereas those responses without any determined cause are known as Non-Specific SCR (NS-SCR) [24]. In this study, the phasic component has a special importance due to its high relationship with the EEG signals, in particular with the delta component, as we will show in Section 3.

#### 2.1.3. EEG Signals

Electroencephalography (EEG) reflects the electrical activity of the brain caused by the synchronized activity of thousands of neurons. Typically, the EEG signal is divided in alpha (α), beta (β), theta (θ), gamma (γ), and delta (δ) waves, which have different frequencies and amplitudes. Commonly, high frequency and low amplitude waves (>7 Hz) such as alpha, beta and gamma are related to the high brain activity or awake state of a person. Alpha waves, with the lowest frequency range (7–12 Hz) for an awake person, are associated to relaxation states, presenting amplitude increments when the eyes are closed. Beta waves (12–30 Hz) have the highest frequency and lowest amplitude in awake state. The beta amplitudes increase when a person plans or execute movements in any body part or when the person observes someone doing any movement (mirror neuron system) [25,26]. Gamma waves are related to rapid eye movements (micro-saccades) which have been associated to information uptake [27], learning, and memory. On the other hand, we find theta and delta waves, which commonly are classified as low frequency EEG waves (<7 Hz). Theta waves (4–7 Hz) with high amplitudes are correlated with difficult cognitive tasks and memorization, whereas those with low amplitudes are associated to a lack of alertness and presence of drowsiness [5]. Delta waves (1–4 Hz) have typically been correlated to deep sleep conditions. However, some results have demonstrated that these kinds of waves are also associated to memory tasks requiring high concentration levels [1,28].

### 2.2. Performed Tasks

#### 2.2.1. EAT (Error Awareness Task)

This task takes 5 min. During this time, a sequence of images with the name of colors written in colored letters are presented. Each image is visible during 900 ms and the interval between two different images is 600 ms. Participants have to press a button (‘Go’ trials) when the color of the letters and the word match (e.g., the word Green appears with green letters). On the other hand, when the color of the letters does not match with the word or when the same word appears in two consecutive trials, the participant avoids pressing the button (‘No Go’ trials) [9,29]. The two ‘No Go’ trials allow the participants to be more attentive than repetitive, especially in the second case, in which, according to [30], individuals are submitted to a stressful situation that increases the alertness condition.

#### 2.2.2. N-Back

This is a memory task that takes 10 min. In this case, tones with different frequencies and duration, normally separated by 3 s, are played through two speakers located in front of the participant. Using pre-determined keyboard computer keys, the user has to identify whether a tone is identical to the n-previous one or not. The number of the n different tones between two similar ones is incremented or decremented depending on the participant performance. The use of tones (auditory stimulus) can be combined with visual stimuli, which are presented simultaneously to increase the complexity [31]. This task is considered as a highly demanding working memory task [32].

#### 2.2.3. Ship Search

This is a task with 20 min of duration in which the participants are asked to identify the moment (using the space bar) and position (specifying the coordinates verbally) of a ship’s appearance in an interactive screen that simulates the water view from a periscope [31]. This kind of task is used to analyze the spatial and temporal awareness of individuals.

#### 2.2.4. PVT (Psycho-Motor Vigilance Task)

In this task, which takes 10 min, participants are asked to click, as soon as possible, the left mouse button when they observe a number on the screen that appears in intervals between 2 and 10 s. The PVT task is performed by every participant in the same computer and using the freely available software proposed in [33,34]. This task has been widely used to analyze the degradation of attention under sleep deprivation, measuring the reaction time (RT) to repetitive stimuli.

### 2.3. EEG and EDA Data Processing

EEG and EDA devices were connected to each participant five minutes before the test. For the EEG case, a cap with ten electrodes was attached to the individual’s head, two used as reference and located on the ears, and the other nine in frontal (F), frontal-polar (Fp), temporal (T), parietal (P), and occipital (O) positions to capture the EEG channels $F_{p2},F_{7},F_{8},O_{1},O_{z},P_{z},O_{2},T_{7}$ and $T_{8}$, as shown in Figure 2. EEG signals were sampled to 200 Hz and filtered between 0.5 and 50 Hz. For the EEG electrodes, their impedance is assured to be lower than 5 K Ω for a good contact with the scalp, and their conductance is increased by means of an electrode gel.

The five EEG bands shown in Figure 3 were obtained using FIR filters designed using the Parks–McClellan optimal equiripple approach.

On the other hand, the EDA signal was obtained using stainless steel electrodes placed on the middle and index fingers of each individual’s non-dominant hand. This signal was decomposed in tonic and phasic components using the convex optimization approach proposed in [35]. The results for an individual executing EAT are shown in Figure 4.

### 2.4. Mutual Information between EDA and EEG

In order to obtain the probability distributions required for the information theory analysis, we discretized the EDA and EEG signals using a uniform count binning process to maximize the entropy and therefore the available information between the studied signals (e.g., the mutual information between the EEG and phasic EDA component), avoiding probability distribution assumptions. In this sense, by means of the Neuroscience Information Theory Toolbox proposed in [15], we defined 12 uniform count bins or states within which the continuous signal values for EDA and EGG were assigned (the value of 12 for the uniform count bins was experimentally chosen between multiple tests. This value offers the best trade between computational requirements (especially simulation time) and acceptable values of entropy and mutual information). This process is described in Figure 5 for the phasic and delta components of EDA and EEG signals, respectively. Observe how, due to the data density differences, some bins were narrower than others in spite of having the same number of observations (2000 observations for each bin). The entropy maximization generates a uniform probability distribution for each state, which is calculated with the expression
(4)$$p\left(s\right)=\frac{N(s)}{N_{obs}},$$
where N(s) is the number of observations classified in a specific bin, and $N_{obs}$ is the total number of observations. In our case, the probability for each state belonging to any EDA or EEG component is $=\frac{2000}{12(2000)}=0.0833$. The joint probability distribution, necessary to calculate the mutual information, was obtained through the cumulative number of times that data within two bins of two different signals appear at the same time. This is described in Table 1 and Table 2 for the case of the phasic component of the EDA signal and the delta component of the EEG signal. Table 1 shows the cumulative number of times that data belonging to any bin of the phasic component and data belonging to any bin of the delta component appear together. On the other hand, Table 2 shows the corresponding joint probability distribution calculated from the cumulative data. Observe how the highest probability values corresponded to the highest cumulative values. Once the joint distribution was obtained, we could apply (Equation 3) to obtain the mutual information between all the EEG components and the phasic and tonic EDA components. The results, shown in Section 3, demonstrate that the highest level of mutual information occurs between the EDA signal and delta EEG component.

## 3. Results

As we have mentioned, the mutual information was calculated between the EDA components (phasic and tonic), and all the five components of EEG (α,β,θ,δ,γ). Additionally, we calculated this measure for all the EEG channels considered in this study ($F_{p2},F_{7},F_{8},O_{1},O_{z},P_{z},O_{2},T_{7}$, $T_{8}$), and for all the tasks described in Section 2.2. Initially, we calculated the mutual information between EDA signal and EEG components, whose results are shown in Table 3. In this case, the cumulative mutual information value was taken for the 12 trials of participant 10 executing the N-Back task. Notice that the EDA signal exhibited the highest mutual information value with respect to the delta component, whereas it was at the minimum for the gamma component. This result was the same when we decomposed the EDA signal in phasic and tonic components, as depicted in Figure 6. In this case, the delta wave was still producing the highest mutual information, with a higher value for the phasic than for the tonic component.

In the case of the EEG channels, as shown in Table 4, the mutual information tended to be higher in frontal regions ($F_{p2},F_{7},F_{8}$) than in the occipital ones ($O_{1},O_{z}$).

In Figure 7, we present the results of the mutual information between EDA and EEG signals for the four tasks described in Section 2.2. These results consider the averaged data of the ten participants, the nine EEG channels ($F_{p2},F_{7},F_{8},O_{1},O_{z},P_{z},O_{2},T_{7},T_{8}$), and the five EEG components (α, β, θ, δ and γ). It is noticeable that the mutual information exhibited the highest values for the delta component, and the lowest values for beta and gamma components for all the tasks; in other words, the lower the EEG component frequency, the higher its mutual information with EDA. Additionally, the task presenting the highest mutual information between EDA and EEG is the N-Back, whereas PVT and EAT have the lowest values. The theta component presented the second highest mutual information values with EDA, with low variations for most of the EEG channels in all trials, especially for EAT and PVT. The alpha component was also present with low mutual information values in some tasks such as the EAT and ship search, especially in the last trials, when the sleep deprivation began to affect the participants’ reaction capability. A considerable reduction in the mutual information was present in the last trials for most of the tasks, especially in the ship search and EAT. In terms of EEG channels, the mutual information presented more activity in frontal regions for the initial trials, and begins to distribute between occipital, frontal and temporal zones as the number of trials increases.

In order to highlight the the brain activity during the trials, Figure 8 shows the normalized mutual information between EEG waves and EDA on each EEG channel and for all the analyzed tasks. If we consider the low frequency EEG waves, we can observe that for the initial trials, $F_{P2}$ is the channel with the highest mutual information for almost all the tasks. On the other hand, channels $F_{7},F_{8},O_{1},O_{2}$ and $O_{Z}$ have considerable values for the last trials, i.e., the mutual information propagates from frontal to occipital, and in some cases to parietal and temporal regions when the sleepiness is present. For the high frequency EEG waves (beta and gamma), mutual information values are less common in frontal regions and more frequent in channels $T_{7},T_{8},O_{1}$ and $O_{2}$; in other words, the temporal activity is greater in higher EEG frequencies vs. lower frequencies.

Finally, in Figure 9, we show the mutual information on each EEG channel in all the tasks considering the contribution of the whole set of trials. In Figure 9a, which shows the results for the phasic component of EDA, we can observe that for the lowest EEG frequencies, the mutual information value is higher for $F_{p2}$ and $F_{7}$ in almost all the tasks, especially for EAT. In contrast, for the highest EEG frequencies (beta and gamma), channels $T_{8},O_{1}$ and $O_{2}$ have considerable mutual information values, whereas channel $P_{Z}$ presents the lowest values, particularly in the ship search task. In the case of the tonic component, which is shown in Figure 9b, in spite of the fact that mutual information values are lower, the behavior is quite similar to that described in the phasic case, except for the remarked low participation of $P_{Z}$ channel in the ship search task.

## 4. Discussion

Results illustrated in Figure 7 demonstrate that the mutual information was the highest between the EDA and delta EEG component for all the tasks, especially in initial trials and for frontal brain region. However, this measure tends to decrease with the number of trials and to propagate from the frontal to other brain zones such as occipital, parietal and temporal zones, which can be attributed to the sleep deprivation effect.

In the case of the N-Back task (Figure 7a), the mutual information values (more than 1 bit) were higher than in the other evaluated tasks. This supposes a strong relationship between the EDA signal and delta component when individuals perform demanding working memory tasks. This result can be contrasted with results described in [1], where a working memory task such as the Sternberg task, demonstrated an increase in delta activity, mainly in frontal lobes. This last aspect, which has also been proposed in [32,36,37,38,39], is equally visible in Figure 7a, where the mutual information in channels $F_{p2}$ and $F_{7}$ is higher for the initial trials, and tends to be uniform for the nine EEG channels as the number of trials progresses.

Figure 7b shows the results for the ship search task, which had the second highest levels of mutual information between EDA and EEG (around 0.5 bits). As described in [1], this value can be attributed to the increase in delta oscillations when sensory afferences are inhibited in activities requiring concentration. In terms of the EEG channels, the ship search exhibited a similar behavior as N-Back, with high mutual information in frontal lobes through channels $F_{p2}$, $F_{7}$ and $F_{8}$ in the initial trials, with an increasing participation of occipital and temporal lobes as the number of trials increases. Additionally, an increase in mutual information between the alpha component and EDA signal for the last trials is noticeable, which can be associated with the inhibition and attention roles that, as mentioned in [2,40], were present in cognitive tasks involving spatial and temporal orientation.

The mutual information between alpha component and EDA signal is also relevant for EAT (Figure 7c). In this case, the inhibitory effect is given by the ‘No-Go’ trials. In terms of the mutual information range, EAT has the lowest values in comparison with the other tasks (less than 0.35 bits), which supposes a weak relationship between the EDA signal and delta component when individuals are facing stressful situations as given by the ‘Go’ and ’No-Go’ trials. The theta component also has a considerable mutual information with EDA for this task in the last trials, when the sleep deprivation begins to affect the participants. This result can be contrasted with the results in [5], where, through a spectral analysis, an association between the individual’s voluntary repression (No-Go trial) and the power amplitude of the theta component is found in a sleep deprivation condition. In relation to the mutual information values for the nine EEG channels, this task exhibits higher values for $F_{p2}$ and $F_{7}$ at the beginning, when individuals are not facing the effects of sleep deprivation.

For the PVT case (Figure 7d), the mutual information values for occipital, parietal, and temporal lobes are higher for the last trials than in the other tasks, especially in channels $P_{z},O_{2},T_{7}$ and $T_{8}$, which can be associated with the results in [7,10], where the delta and theta power increases in occipital areas when the response time (RT) of PVT task begins to increase as a consequence of sleep deprivation. Again, we can observe in Figure 7d that the mutual information becomes more uniform with the number of trials for all the EEG channels, although, in relation to the other tasks, it does not decrease. This case corresponds with the results found through the power spectral analysis in [6,7], where the presence of alpha, delta, and theta components increases with sleep deprivation and loss of the vigilance ability.

In Figure 8, we show the normalized mutual information in order to highlight the EEG channels’ behavior during the 12 trials and the 4 analyzed tasks. For the N-Back task case, described in Figure 8a, we can observe a prominent mutual information in frontal zones for low EEG frequencies (delta, theta and alpha) that diminishes and translates to the occipital regions ($O_{1},O_{2}$) in the last trials. In the high frequency cases (beta and gamma), parietal and temporal regions have considerable values, which also decrease with the number of trials. These results are similar to the results shown in [41,42,43], where, through a power spectral analysis, authors demonstrate that the frontal theta power increases and parietal beta power decreases with working memory complexity, which can be comparable with the difficulty to perform the task in sleepiness conditions. The ship search task results are shown in Figure 8b. In this case, the frontal activity does not attenuate for theta and alpha components as in the N-Back task case, which, as we have mentioned before, can be attributed to the inhibitory and attentive roles involved in spatial and temporal activities described in [2,40]. In addition, an important activity is distinguishable in the right temporal lobe for beta and gamma waves, which can reflect the spatial awareness required in these types of tasks [44]. In the EAT case (Figure 8c), the frontal activity is the most common for all the EEG waves. This effect can be associated to the theta and alpha power increase in this brain region due to the voluntary repression (No-Go trial) or the inhibitory effect that these kinds of tasks imply in sleep deprivation conditions [2,5]. In Figure 8d, we reaffirm that PVT presents relevant activity in the occipital region as a consequence of the response time increase caused by sleepiness, which can be contrasted with results in [7,10].

Finally, in Figure 9, we have decomposed the EDA signal in phasic and tonic components to find their normalized mutual information with each EEG component and each EEG channel for all the analyzed tasks. In this case, not only the mutual information values obtained for the 10 participants were averaged, but also the values obtained during the 12 trials. It is evident that the frontal region is the most relevant for all the studied tasks in terms of mutual information between EDA (tonic and phasic) and low frequency EEG waves (delta, theta and alpha). Beta and gamma also have frontal activity, especially for the N-Back (working memory task). Again, we can observe right temporal activity for the ship search task, and relevant mutual information in the occipital region for PVT, which corresponds with the previous analysis. A comparison between phasic and tonic results shows that the tonic analysis can provide additional information for high EEG frequencies.

## 5. Limitations and Future Directions

Power Analysis. We run a power analysis using as the effect size the Pearson’s correlation to obtain the *p*-values shown in Table 5. Although these *p*-values are low enough to consider a statistical relationship between the EEG and EDA measures (null-hypothesis rejection), especially for the phasic case, the lowest sample size (theta-phasic case) required to obtain a considerable statistical power was of 219 individuals. However, due to the rigorous 25-h sleep deprivation experiment with demanding protocols both for the conductors and subjects, and the lack of individuals disposed to face sleep deprivation for such an extended period, this study was focused on a narrow group of 10 young and healthy subjects, as we have mentioned before. In this sense, the limited sample size and the specific characteristics of the subjects make the observations restricted to this particular group. To reach broader conclusions, further data collection is necessary.

Heart Rate Variability (HRV). In order to assess variations in parasympathetic activity throughout the experiment, we employed the RMSSD measure to calculate HRV. The results are presented in Figure 10 and Figure 11. Figure 10a demonstrates short RR-intervals (around 700 ms) after 10 h from the start of the experiment, indicating reduced parasympathetic activity during the night hours caused by sleep deprivation. In contrast, Figure 10b reveals an increase in RR-intervals (around 950 ms) during the final trials conducted during the early morning hours, a result that we plan to investigate in future studies. On the other hand, Figure 11 displays the overall HRV variations across all analyzed tasks. Notably, a reduction in this parameter is observed for trials 5 to 8 in all cases, coinciding with the night hours (from 20 h to 24 h).

## 6. Conclusions

In this work, we have found the mutual information between EDA and EEG in order to include linear and non-linear interactions. The signals were taken from ten participants developing four cognitive tasks (EAT, ship search, PVT, and N-Back) during 12 trials developed in 24 h. The results demonstrate that this measure is higher between the delta EEG component and EDA, especially with its phasic component, and lower, in the same order, for theta, alpha, gamma and beta, i.e., the lower the EEG wave frequency, the higher its mutual information with EDA. For the four analyzed tasks, this measure is higher for N-Back and lower, in the same order, for the ship search, PVT and EAT. This means that the EDA signal can be especially used to analyze the performance in working memory tasks, where delta oscillations are significative. On the other hand, the mutual information between the EDA (particularly for tonic component) and alpha EEG component demonstrates an important increase in the number of trials in tasks requiring inhibition and attention behaviors, especially in the ship search and EAT, which can be associated with other results found in literature that use a spectral analysis and other statistical methods, such as ANOVA. Additionally, an increase in the mutual information between the EDA and theta component is also visible with the number of trials, which can be associated with the fatigue caused by sleep deprivation and response time delays described in other approaches using different techniques [40,45,46,47].

In terms of the nine EEG analyzed channels, the mutual information values, for the initial trials, are high for channels $F_{p2},F_{7}$ and $F_{8}$, and tend to be uniform and reduced for all the tasks, because the number of trials increases, in other words, as the sleepiness appears. These results can be contrasted with results found in some approaches using different techniques, in which a relevant role of frontal lobes in cognitive functions such as attention, working memory, decision making and inhibitory control is described. For the PVT case, there is also a relationship with the findings of other approaches, where an increment in delta and theta power is demonstrated for occipital areas as a consequence of sleep deprivation. For the case of ship search, a particular behavior for the mutual information is described in the right temporal region and for the gamma wave, which can be related to the spatial awareness involved in this task.

In general, the mutual information between EDA and EEG signals, and its correspondence with important results found in the literature based exclusively on the EEG signal, suggests that exploring EDA could be an intriguing alternative for investigating brain behavior.

## Figures and Tables

**Figure 1 behavsci-13-00707-f001:**
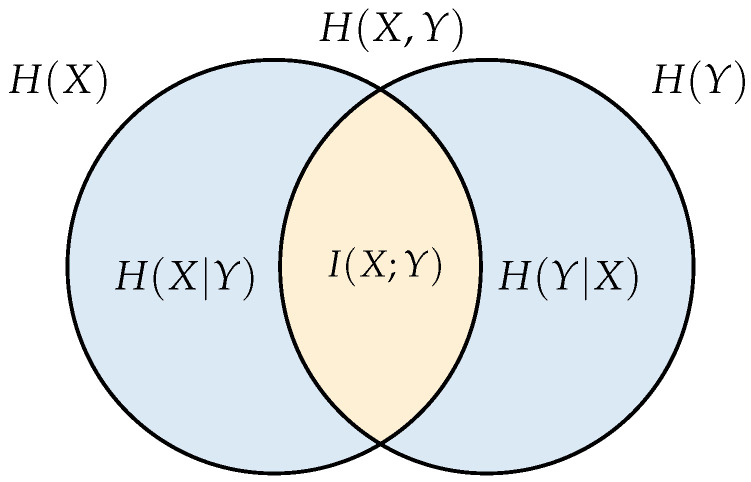
Information theory measures.

**Figure 2 behavsci-13-00707-f002:**
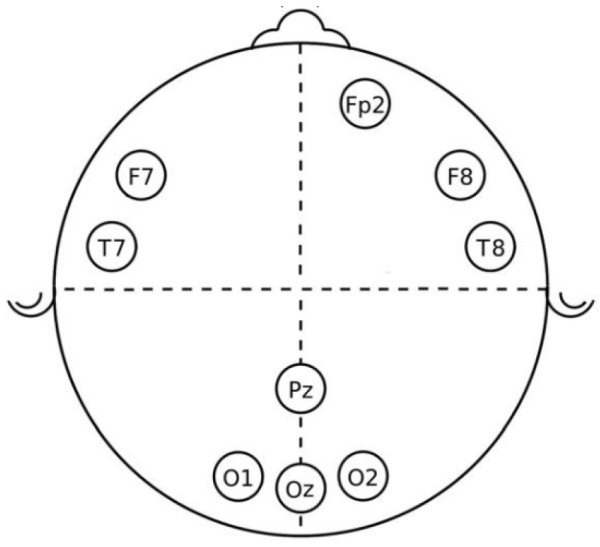
EEG channels.

**Figure 3 behavsci-13-00707-f003:**
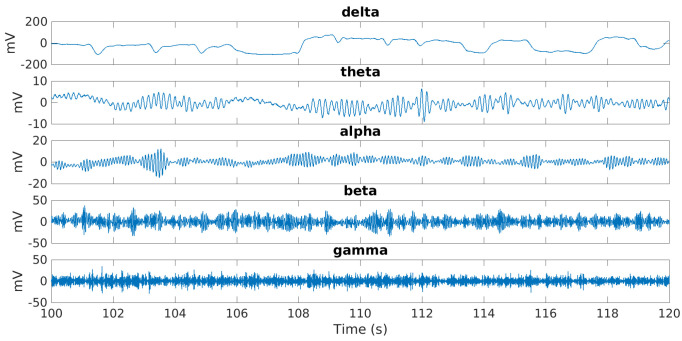
EEG signals.

**Figure 4 behavsci-13-00707-f004:**
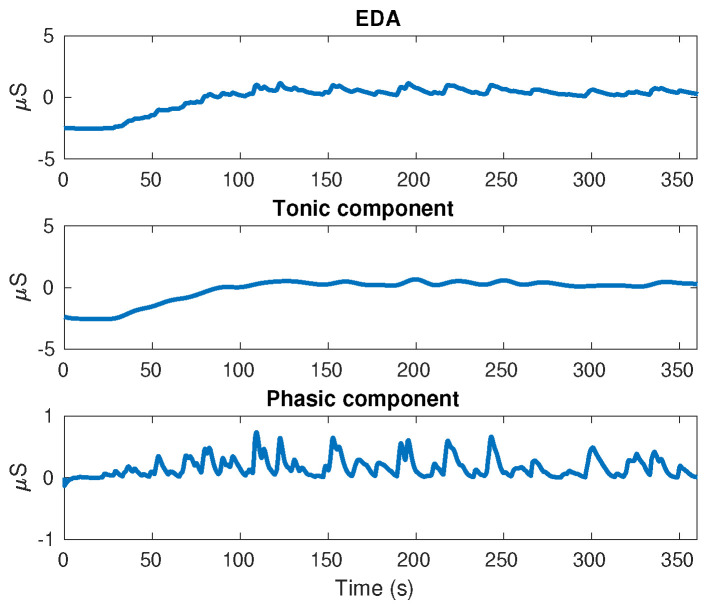
EDA and its components, Tonic and Phasic.

**Figure 5 behavsci-13-00707-f005:**
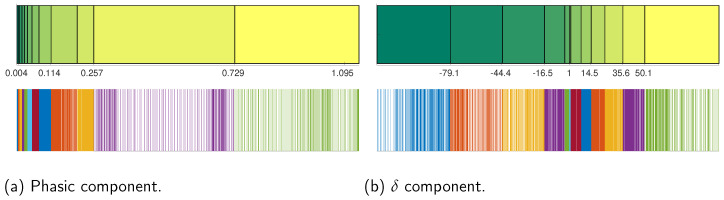
Binning process for phasic (**a**) and δ (**b**) components. Above are the 12 bins with their corresponding sizes and value limits. Below is the data (2000 observations) distribution within each bin.

**Figure 6 behavsci-13-00707-f006:**
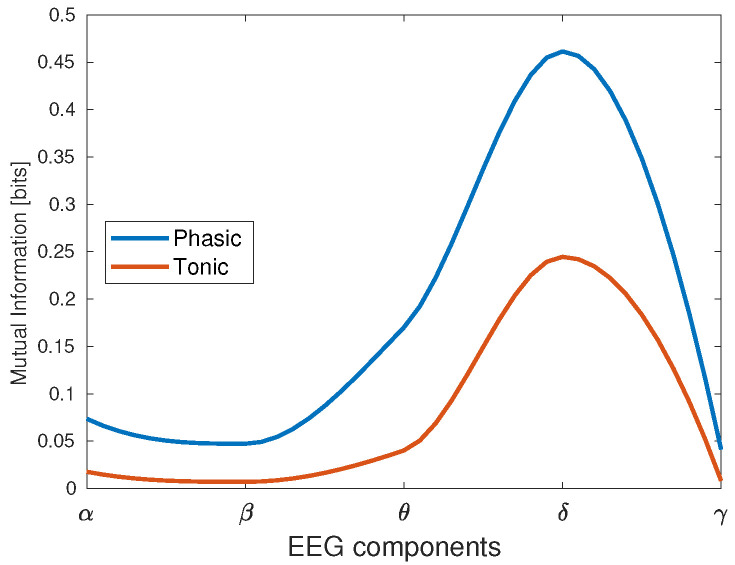
Mutual information between tonic and phasic EDA components vs. EEG signals.

**Figure 7 behavsci-13-00707-f007:**
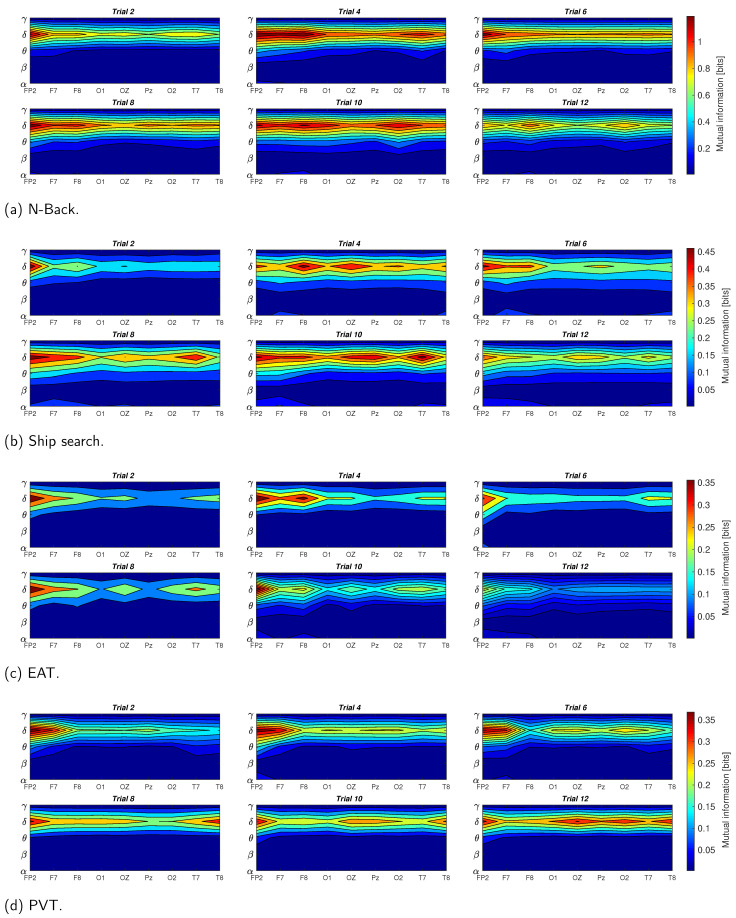
Mutual information between EDA and EEG components (α,δ,θ,β,γ) for all the EEG channels, and all the analyzed tasks. (**a**) N-Back. (**b**) Ship search. (**c**) EAT and (**d**) PVT. Results were averaged for all participants. In order to avoid redundancy, only even trials are shown. The mutual information is the highest for N-Back (greater than 1 bit) and the lowest for PVT (lower than 0.35 bits).

**Figure 8 behavsci-13-00707-f008:**
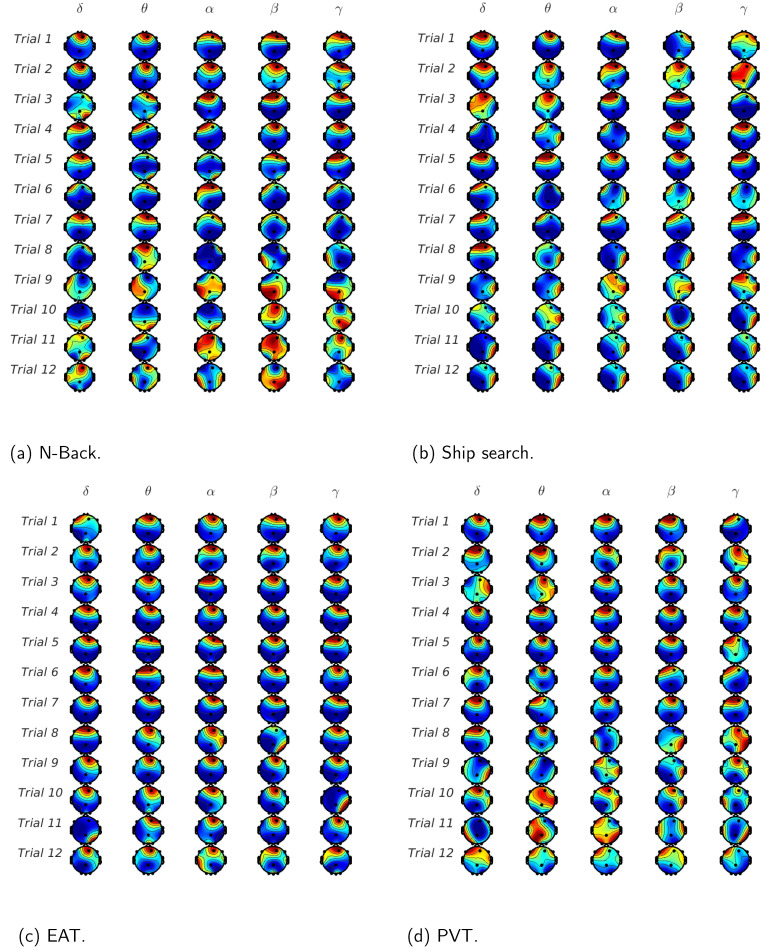
Scalp topographies of normalized mutual information between EDA and EEG components (α,δ,θ,β,γ) for all the analyzed tasks. Results were averaged for all participants.

**Figure 9 behavsci-13-00707-f009:**
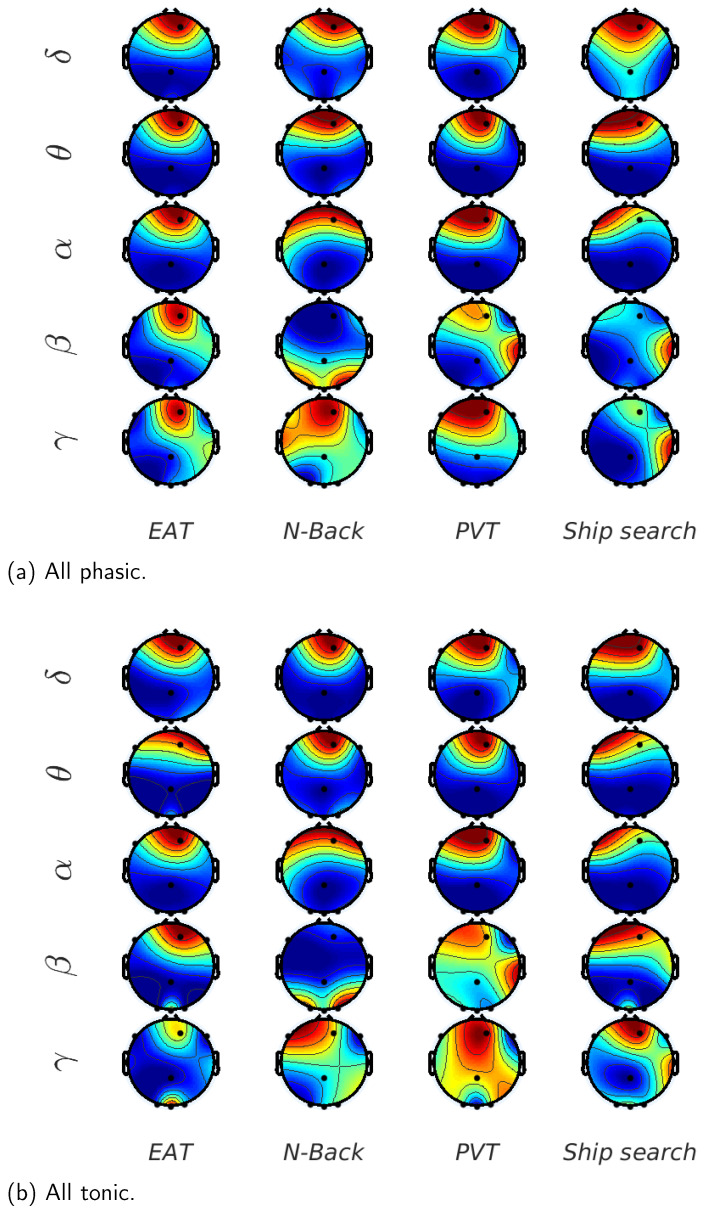
Scalp topographies of normalized mutual information between EDA and EEG components (α,δ,θ,β,γ) for all the analyzed tasks. (**a**) Phasic component. (**b**) Tonic component. Results were averaged for all trials and participants.

**Figure 10 behavsci-13-00707-f010:**
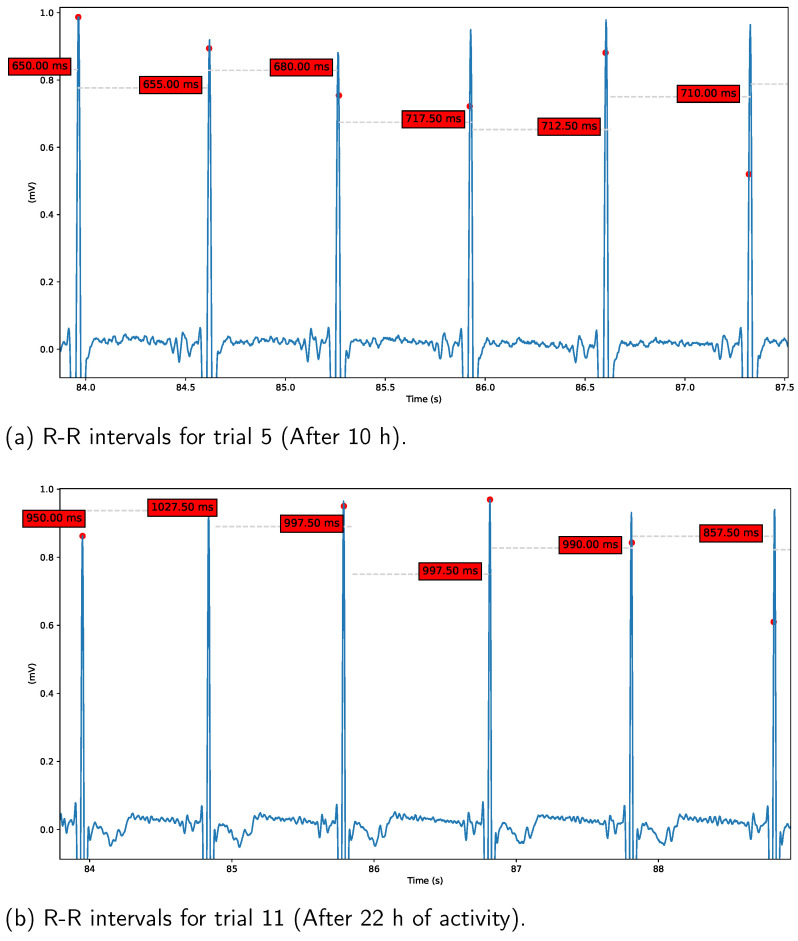
RR-intervals after 5 trials (taken around 20:00 h) (**a**), and RR-intervals for early morning hours of the second day (trial 12) (**b**). A decrease in RR-intervals is exhibited around 20:00 h, which can be associated to parasympathetic activity reduction because of the beginning of normal sleeping hours.

**Figure 11 behavsci-13-00707-f011:**
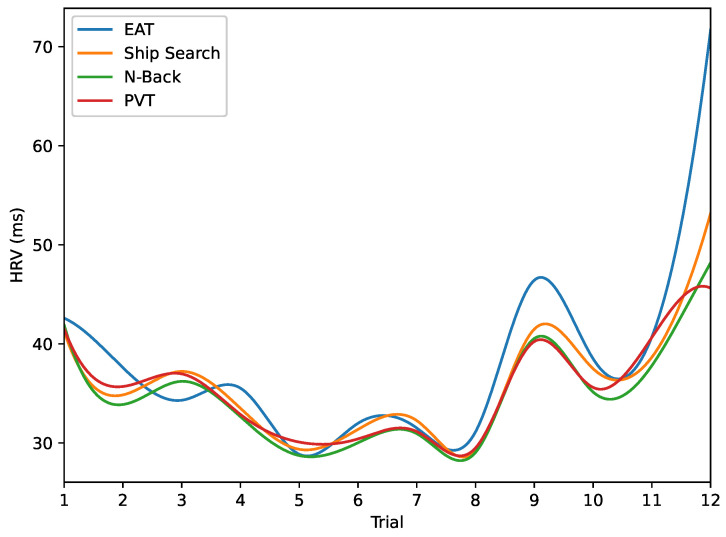
Average HRVs for the analyzed tasks. HRV values decrease from trials 5 to 8 for all activities (from 20 h to 24 h).

**Table 1 behavsci-13-00707-t001:** Cumulative number for δ and phasic bins.

δ Bins								
		**1**	**2**	**3**	**4**	**5**	**.**	**12**
Phasic bins	1	64	118	322	254	407	.	97
	2	98	416	201	84	169	.	282
	3	99	250	105	132	177	.	139
	4	81	224	102	32	97	.	94
	5	0	60	119	117	216	.	114
	.	.	.	.	.	.	.	.
	12	1219	250	116	105	75	.	0

**Table 2 behavsci-13-00707-t002:** Joint probability distribution between phasic and δ bins.

δ Bins								
		**1**	**2**	**3**	**4**	**5**	**.**	**12**
Phasic bins	1	0.0027	0.0049	0.0134	0.0106	0.0170	.	0.0040
	2	0.0041	0.0173	0.0084	0.0035	0.0070	.	0.0117
	3	0.0041	0.0104	0.0044	0.0055	0.0074	.	0.0058
	4	0.0034	0.0093	0.0042	0.0013	0.0040	.	0.0039
	5	0	0.0025	0.0050	0.0049	0.0090	.	0.0047
	.	.	.	.	.	.	.	.
	12	0.0508	0.0104	0.0048	0.0044	0.0031	.	0

**Table 3 behavsci-13-00707-t003:** Mutual information between EEG components and EDA (N-Back task, participant 10, 12 trials, and all EEG bands).

EEG Component	MI with EDA [bits]
α	7.5386
β	4.6661
θ	8.9486
δ	63.0043
γ	3.3007

**Table 4 behavsci-13-00707-t004:** Mutual information between EEG components and EDA (N-Back task, participant 10, 12 trials, and all EEG channels).

EEG Channel	MI with EDA [bits]
FP2	11.5861
F7	10.5022
F8	9.8471
O1	8.9789
OZ	9.0073
Pz	9.3320
O2	9.1991
T7	9.3958
T8	9.6097

**Table 5 behavsci-13-00707-t005:** Correlation Coefficients and *p*-values.

EEG-Band	EDA-Component	Pearson’s Correlation	*p*-Value
Alpha	Phasic	−0.188	$3.80^{-189}$
Beta	Phasic	0.086	$5.16^{-41}$
Theta	Phasic	0.268	0.00
Delta	Phasic	0.238	$6.62^{-306}$
Gamma	Phasic	0.004	$5.19^{-1}$
Alpha	Tonic	−0.070	$1.17^{-27}$
Beta	Tonic	0.031	$1.69^{-6}$
Theta	Tonic	0.100	$3.33^{-54}$
Delta	Tonic	−0.301	0.00
Gamma	Tonic	0.003	$6.88^{-1}$

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
