# Peer review of "Mutual Information between EDA and EEG in Multiple Cognitive Tasks and Sleep Deprivation Conditions"

_behavsci, 2023, doi:10.3390/bs13090707_

Round 1
Reviewer 1 Report (Previous Reviewer 1)
General statement
The article titled "Mutual Information Between EDA and EEG in Multiple Cognitive Tasks and Sleep Deprivation Conditions" explores the relationship between electrodermal activity (EDA), electroencephalography (EEG), cognitive tasks, and sleep deprivation. While the topic is intriguing, the author's responses and modifications during the second-round review are not entirely satisfactory. Several critical points need to be addressed to improve the paper's quality and clarity.
Point 1
The use of sleep deprivation protocols in research is undoubtedly challenging for both researchers and participants. However, this should not be used as an excuse for collecting data from a small sample size of only ten subjects. The authors themselves acknowledge the need for further data collection, which suggests that they should gather more data before submitting the article for review. Moreover, it is inappropriate to discuss the sample size issue within the results section. The sample size and data collection limitations should be addressed separately in the methodology or limitations section.
Point 2
Control variables play a crucial role in research, particularly in studies involving sleep deprivation. The paper lacks a proper comparison between the sleep deprivation condition and either a well-defined sleep condition or a baseline. This omission raises concerns about the validity of the findings and the effectiveness of the sleep deprivation manipulation. The authors mention heart rate variability (HRV) as a possible control variable for assessing sleep deprivation manipulation, but they also treat it as a variable of interest. They stated “We also analyze ECG signals taken on each trial to study the HRV.” And “Additionally, ECG signals were collected during each trial to analyze the effect of sleep deprivation in HRV.” This creates ambiguity in the study design, and the authors must clarify and demonstrate that their sleep deprivation protocol effectively deprived participants of sleep.
Point 3
Authors do not explain the rationale behind the chosen task order. The absence of information on why tasks were not randomized across participants is a notable flaw in the experimental design. Randomization is crucial to eliminate potential confounding variables and ensure unbiased results. The authors need to provide a clear explanation for their task order selection or justify why randomization was not feasible or appropriate.
Point 4
The results section still lacks essential details, such as reaction times and accuracy, which are fundamental aspects of cognitive task performance. The author's response to this concern does not address the issue satisfactorily and appears to be out of context. Including these critical performance metrics is essential to allow readers to assess the cognitive effects of sleep deprivation accurately.
Point 6
These references are not enough to describe their tasks. The paper still lacks sufficient references and details to adequately describe the cognitive tasks employed in the study. Information on the number of trials included, task timings, and other relevant specifics is crucial for readers to comprehend the experimental procedures fully. The authors should provide comprehensive task descriptions, including trial counts and timing details, to enhance the reproducibility and transparency of their research.
This paper requires more clarity regarding its hypothesis and methods. Ambiguities in the research design, data collection, and analytical approach need to be addressed to enhance the overall quality of the article. A more thorough and transparent presentation of the research methodology and results will make the paper more accessible and credible to the scientific community.
This paper still needs proof reading
Author Response
Please see the attachment.

Reviewer 2 Report (Previous Reviewer 2)
The authors have addressed my queries. I have no further queries/concerns.
Author Response
The reviewer has not more comments because the queries were addressed in round 1.
Reviewer 3 Report (New Reviewer)
The manuscript is very interesting and well-written. The authors aimed to study the mutual information between EDA and EEG in cognitive tasks under sleep deprivation conditions.
1. My first question is regarding the study participants' exclusion criteria and were the subjects screened for sleep disorders?
2. As sleep deprivation was included in the design it’s very important to have precise information on study participants’ sleep-wake habits before the study. Did you monitor the sleep-wake behavior of study participants before the study?
3. Was the subjects' chronotype identified?
4. Why the authors used an EEG montage with just one reference electrode?
5. Was the very slow activity, lower than 1 Hz oscillations, analyzed separately?
6. pg 15 lines 397-399 “In general, the results show correspondence with many important results found in the literature that use EEG analysis, which makes this approach an interesting alternative to analyze brain activity in a less complex form as EDA analysis suggests.” - statement needs reformulation.
7. Reference 1 and 33 are duplicated in the References.
Author Response
Please see the attachment.

Reviewer 4 Report (New Reviewer)
The article entitled as Mutual Information Between EDA and EEG in Multiple Cognitive Tasks and Sleep Deprivation Conditions by David Alejandro Martínez Vásquez et al is very interesting .The article is very good however a few comments need clearance and justification the authors:
1. add headings to the abstract section.
2. What instrument you used for EEG?
3. Please concise the conclusion and add limitations and future perspective
4. Recheck grammar and remove typo mistakes.
Minor grammatical errors
Author Response
Please see the attachment.

This manuscript is a resubmission of an earlier submission. The following is a list of the peer review reports and author responses from that submission.
Round 1
Reviewer 1 Report
This review is for the article titled, “Mutual Information Between EDA and EEG in Multiple Cognitive Tasks and Sleep Deprivation Conditions”. The authors investigated an interesting topic, the effects of sleep deprivation on the relationship between central and sympathetic nervous system.
The manuscript tackles an intriguing question, however there are major issues.
Sample size of ten participants is too small. It is generally advised to run a power analysis to know your sample size. Moreover, sleep quality and circadian rhythms of participants were not controlled. They are known to affect results.
It is important to show that sleep deprivation manipulation was effective, like with subjective questionnaires or EEG signal.
Participants performed tasks battery every two hours for 25 hours. In the result section, it is unclear which data were analyzed, at what timing.
Tasks behavioral results were not reported.
The tasks battery lasted minimum 45 min. It means that participants had one hour to rest and then tested again. Participants might have been exhausted only by repeating tasks battery so many times. I am wondering why you did it, especially if in the results section you do not show results along the night of sleep deprivation.
References should be added per each task used. It is unclear how these tasks were built. The EAT task sounds like the Stroop Task. And the N-Back task seems not conformed to the classical version: it is written “The number of the n different tones between two similar ones is incremented or decremented depending on the participant performance” lines 166-168. Usually, the rule for the n-back (0-back, 1-back, 2-back or 3-back) it is set at the beginning of the task or each block. If the rule changed based on performance, how did they know which rule they had to follow?
Based on all points mentioned above, results and conclusions are questionable.
Finally, i suggest to rewrite and reorganize your paper. Here are some suggestions. Specifically, rewrite introduction and methods: lines from 71 to 84 should not be in the Introduction; Section 2 preliminaries should be moved into Methods or just mention briefly in Methods.
I suggest proof-reading.
Reviewer 2 Report
In the current study, the authors calculated mutual information between EDA and EEG activity to consider linear and non-linear interactions, which could shed light on the relationship between brain activity and peripheral autonomic sympathetic activity. Overall, the paper is well written, the methods are well described, the statistical methods used are appropriate, and most of the interpretations are acceptable.
I have a couple of primary concerns about the study design. The subjects performed the tasks every 2 hours over a 24-hour period, which I assume was designed to delineate the effect of sleep deprivation on the cognitive tasks.
1. To address the concern about the effect of sleep deprivation on cognitive tasks, the authors could consider including a control group of subjects who are not sleep deprived. This would allow for a direct comparison between the two groups and help to isolate the effects of sleep deprivation on the observed outcomes.
2. Moreover, the time when the trials were conducted must be mentioned. Varying the time of the initial and later trials could also affect the results. It would be helpful for the authors to provide more information about the timing of the trials to ensure that the results are not influenced by any confounding factors.
Overall, to enhance the clarity of the study design, the authors could provide a more detailed explanation of their reasoning behind the decision to conduct the trials every 2 hours over 24 hours. This helps address any potential concerns about the validity of the observations due to sleep deprivation or other factors.
